# Detection of Pilot’s Mental Workload Using a Wireless EEG Headset in Airfield Traffic Pattern Tasks

**DOI:** 10.3390/e25071035

**Published:** 2023-07-10

**Authors:** Chenglin Liu, Chenyang Zhang, Luohao Sun, Kun Liu, Haiyue Liu, Wenbing Zhu, Chaozhe Jiang

**Affiliations:** 1School of Transportation & Logistics, Southwest Jiaotong University, Chengdu 611756, China; lcladress@my.swjtu.edu.cn (C.L.);; 2School of Information Science and Technology, Southwest Jiaotong University, Chengdu 611756, China

**Keywords:** mental workload, flight safety, system, EEG, wireless headset

## Abstract

Elevated mental workload (MWL) experienced by pilots can result in increased reaction times or incorrect actions, potentially compromising flight safety. This study aims to develop a functional system to assist administrators in identifying and detecting pilots’ real-time MWL and evaluate its effectiveness using designed airfield traffic pattern tasks within a realistic flight simulator. The perceived MWL in various situations was assessed and labeled using NASA Task Load Index (NASA-TLX) scores. Physiological features were then extracted using a fast Fourier transformation with 2-s sliding time windows. Feature selection was conducted by comparing the results of the Kruskal-Wallis (K-W) test and Sequential Forward Floating Selection (SFFS). The results proved that the optimal input was all PSD features. Moreover, the study analyzed the effects of electroencephalography (EEG) features from distinct brain regions and PSD changes across different MWL levels to further assess the proposed system’s performance. A 10-fold cross-validation was performed on six classifiers, and the optimal accuracy of 87.57% was attained using a multi-class K-Nearest Neighbor (KNN) classifier for classifying different MWL levels. The findings indicate that the wireless headset-based system is reliable and feasible. Consequently, numerous wireless EEG device-based systems can be developed for application in diverse real-driving scenarios. Additionally, the current system contributes to future research on actual flight conditions.

## 1. Introduction

Although automated operations and artificial intelligence have been extensively developed to assist in flying, pilots still play a critical role in ensuring system safety for possible system malfunctions and operational limitations, particularly in dangerous flight situations that can suddenly cause excessive mental workload (MWL) [1,2]. Mental workload, for instance, is a transactional process that illustrates the interaction between an individual’s abilities (finite information processing capacity and variable cognitive demands) and a given task’s requirements [3]. In civil aviation, decades of flight accident statistics indicate that excessive mental stress is the leading cause of aircraft accidents [4]. Consequently, detecting pilots’ real-time MWL is crucial for administrators to evaluate pilots’ situation awareness (SA) and guide them to allocate their attention reasonably, thereby improving decision quality and reducing human error. Identifying a potential increase in MWL and subsequently providing system corrective actions by the users has been considered a critical process associated with flight safety. Therefore, an executable and reliable system based on a 5-channel wireless EEG headset (WEEGH) was proposed in this study to assist administrators in detecting pilots’ real-time MWL levels in air traffic control, ensuring flight safety.

In terms of detecting MWL, numerous studies have been conducted across different domains, including navigation, transportation, and aviation, utilizing diverse methods and corresponding experiments. These studies are mainly based on various physiological signals such as electroencephalography (EEG), electrocardiogram (ECG), electromyogram (EMG), and eye-tracking (ET) [5,6]. For instance, Li et al. proposed a two-phase analytical methodology for revealing SA-related neuro-physiological patterns and recognizing air traffic control officers’ (ATCOs’) SA loss using EEG and eye-tracking data [7] and proved that the neuro-physiological behaviors of SA loss during normal workloads differed from those in high workload situations. In Lin et al.’s study [8], the EEG power features were fed into a support vector regression (SVR) model to estimate drivers’ reaction time, which was translated into drivers’ vigilance level. Jiao et al. fed ECG signals into multiple classifiers to research the stress levels of high-speed rail train drivers under various operating conditions, and the result showed that the random forest (RF) model performed the best [9]. Compared with other drivers, aircraft pilots have to operate more complex vehicles and therefore go through a stricter training program before getting their flying license [10]. Although modern cockpits are designed to be as intuitive as possible, a proper and accurate understanding of the relevant information among the many presented in the cockpit is taxing but crucial for pilots, especially in emergencies in which the time available for understanding the problem could be very short. Hence, given the complexity of aircraft systems, proposing a system to identify controllers’ instantaneous MWL is momentous so that preventative measures could be adopted in time when pilots suffer from dangerous MWL situations.

Among all the physiological signals, EEG is regarded as one of the most promising signals for detecting human performance involved in MWL change [11]. EEG refers to electrical activities characterized by rhythmic patterns that are generated by neurons in the brain and detected on the surface of the scalp by placing sensors on specific brain regions like the prefrontal cortex (PFC) [12]. Compared with other neuro-ergonomics techniques, such as functional magnetic resonance imaging (fMRI) and near-infrared spectroscopy (NIRS), EEG signals have a higher temporal resolution; they can reach millisecond or even microsecond levels and carry a lot of real-time biological information [13]. The rhythmic activity of each frequency band has a specific scalp distribution and biological significance. For example, when people are awake and keep their eyes open, alpha waves are usually measured in the occipital lobe of the brain and will weaken with the increase of eye-opening or psychological activity [14]. An increase in EEG power spectra in the band is believed to be related to increased MWL and cognitive demands [15]. In the experiment, sensory stimuli or cognitive tasks can enhance or weaken the rhythmic amplitude of EEG rhythms in specific frequency bands [16], and the amplitude changes of each frequency band can reflect the mental workload status of pilots [17]. Therefore, the following six frequency bands of EEG signals are selected to extract the features for mental workload identification in this study: (1–3 Hz), (4–7 Hz), (8–11 Hz), 1 (12–20 Hz), 2 (21–29 Hz), (30–40 Hz). In addition, the brain’s cerebral cortex can reflect all conceivable physiological responses to any given task instantaneously, thereby offering a more profound insight into the study of human behavior [18]. Different brain regions are responsible for different primary functions. The frontal lobe is mainly involved in some higher-level functions such as decision-making, planning, and problem-solving; the temporal lobe is mainly involved in processing auditory information, language, and memory; and the occipital lobe is primarily responsible for processing visual information [19].

The EEG is a versatile tool that has been widely used to predict an individual’s MWL [10]. It has been shown that EEG features can reflect the state of human beings, and it is also considered one of the most effective physiological signal sources to evaluate the MWL of the human being [20,21]. To analyze mental workload, various EEG features were used based on the characteristics of different bands and brain regions. For example, Yiu et al. [7,22] used the PSD of each band in all brain regions to analyze pilots’ mental workload, while Li et al. [7] utilized 12 calculative features from different domains to explore pilots’ mental workload and its related situation awareness. EEG has been widely used in the field of neuroscience [23]; in addition, with the convergence of various research areas, it has also been gradually used in these fields combined with corresponding tools. So far, many scholars have begun to use EEG signals to study the MWL of drivers.

Recently, technological advancements such as wireless recording, advanced sensor technology, cost-effective amplifiers, real-time temporal resolution, machine learning analysis, and sophisticated signal analysis methodologies have made portable EEG headsets easily accessible to researchers in various scientific fields [24]. In addition, due to the advantages of good non-invasive wireless connectivity, portability, excellent user-friendliness, and strong endurance [25], WEEGHs provide a means of evaluating the brain dynamics involved in the integration of perceptual functions across different scenarios. Currently, some applications in different research sectors already exist (medicine, communication and control, entertainment, rehabilitation, assistive technology, training, and others). For example, Zhang et al. used EEG features acquired by a WEEGH to examine the effects of higher temperature setpoints during the summer on office workers’ cognitive load and thermal comfort. LaRocco et al. presented a systematic review of available low-cost EEG headsets used for drowsiness detection [26]; they evaluated their effectiveness and provided recommendations for their use in detecting drowsiness, which is a leading cause of traffic and industrial accidents. In contrast to multi-channel EEG devices, WEEGHs have the advantages of good non-invasive wireless connectivity, portability, excellent user-friendliness, and strong endurance [25]. Accordingly, some consumer-grade EEG devices should be considered in research in other areas. The real-time detection of pilots’ EEG load has the following advantages with the use of WEEGH: (a) minimizing the additional load brought by wearing EEG devices, improving flight safety and experimental accuracy; (b) meeting the needs of pilots to perform flight tasks for long periods; and (c) helping administrators grasp the pilot’s MWL status in real-time. At the same time, WEEGHs are easily accepted by pilots due to their simple wearing procedures and good comfort, which is extremely important for subsequent real-flight research [27].

Despite the advancements made in developing some methods to detect the MWL levels of pilots, the following three research gaps still exist: (a) to the authors’ knowledge, very few in-depth studies have detected pilots’ real-time MWL using consumer-grade EEG devices. There has been some research in various fields based on portable dry-electrode EEG devices. For instance, Thomas et al. evaluated the technical and practical usability and efficacy of a new portable dry-electrode EEG recorder in patients’ diagnosis and therapy. The results show that despite the slightly worse overall technical signal quality, the WEEGH is well-suited for use in everyday neurological outpatient practice [28]. Payongkit et al. first compared the performances of OpenBCI with those of research-grade systems, employing the same algorithms that were applied to benchmark datasets [29]. The results indicated that the prediction accuracies were decidedly comparable to those of previous studies that utilized more costly EEG amplifiers for data acquisition. Therefore, advancements in sensor and communication technology enable some consumer-category EEG devices to be considered in research detecting pilots’ real-time MWL if the accuracy of the data is positive enough. (b) The massive amount of EEG data collected by research EEG devices contains a significant amount of redundant features, which considerably slow down the computation and processing speed of the EEG signals and consume a considerable amount of computational cost. EEG devices with 32 or 64 channels are usually considered by researchers for their quantity and accuracy of signals [30]. While utilizing additional channels may be appealing and useful in certain scenarios, even for localized coverage, employing more channels will also require more preparation time, be more invasive, mandate increased data storage for multi-channel use, produce redundant data, and increase time and cost for computation and analysis [31]. Therefore, increased electrode quantities in EEG systems do not necessarily equate to improved performance. (c) An absence of a ‘bridge’ between experimentation and practical application for studies based on EEG devices can hardly applicate in real flight tasks. For example, Frédéric et al. monitored pilots’ MWL using a six-dry-electrode EEG system in real flight conditions, but the accuracy of single-trial classification could only achieve 70%. However, this study demonstrated the potential of dry-electrode EEG and revealed that hardware improvement is still needed before it could be used for everyday flight operations [27]. Leandro L. et al. investigated the effects of flight procedures of varied complexity on the in-flight EEG activity of military helicopter pilots in real flight conditions only using analysis of variance (ANOVA) based on a portable EEG recorder. The result indicated that EEG could be used to evaluate online cognitive performance in real-life scenarios. Motivated by these, an online MWL-detecting system employing compatible WEEGHs in aviation would be required, which can be referenced by real flight studies and applied to daily pilots. Before developing an online real-time detection system, the following research questions shall be addressed:R1: Can we determine if the data accuracy collected by this EEG device is adequate for detecting the real-time mental workload of pilots?R2: Is there a suitable model that can identify pilots’ different MWL levels?R3: How can we determine the optimal input among all features?

In this study, an airfield traffic pattern task that included three phases involving different MWL levels was designed based on participants’ daily flight training subjects and corresponding airfields. Following each participant’s experiment, perceived MWL was measured using NASA-TLX for labeling samples’ MWL with low, middle, and high status. Different features from multiple dimensions, including time-domain (mean, variance, standard deviation, peak-to-peak (PTP) amplitude, skewness, kurtosis, and root-mean-squared value), frequency-domain (power spectral density (PSD)), and entropy (sample entropy and approximate entropy), are extracted. The performance of the Kruskal-Wallis (K-W) test and sequential forward floating selection (SFFS) were compared to select the optimal inputs based on the analysis of feature dimension and classification accuracy of all feature sets. The aim was to determine the best method for selecting the most suitable features. Finally, a system for detecting the real-time MWL of pilots is developed based on all results. In summary, the contributions of this paper are described as follows:A real-time detection system for pilots’ MWL based on this WEEGH was proposed. This system had advantages such as low cost, low data redundancy, and fast computational speed. It also provided a direction for studying pilots’ MWL in real flight conditions or other real driving scenarios based on consumer-grade EEG devices.The impacts of different EEG features, based on this system, from different brain regions and different bands on MWL were investigated. The results of this study contribute to understanding the MWL mechanisms of pilots from the perspective of EEG and provide support for feature selection in future MWL-detecting research.

## 2. Materials and Methods

This study proposed a system for detecting pilots’ real-time MWL using EEG data while examining the associated physiological behaviors and the impact of different brain regions (Figure 1). The EEG data were acquired using a wireless EEG headset (WEEGH) and pre-processed into 2-s epochs with a 50% overlap rate. The NASA-TLX scores were employed to label high, medium, and low MWL samples in modified airfield traffic pattern tasks that encompassed different MWL levels. To develop the system, multiple feature sets were chosen using two feature selection algorithms from the annotated EEG data, which were then fed into different classification algorithms to identify the optimal input and algorithm. To understand the physiological patterns and validate the developed system, PSD changes in distinct brain activities and the impact of EEG features from different brain regions were also considered and investigated. Consequently, a potential application was proposed, incorporating an EEG sensor-integrated headset to aid administrators in detecting pilots’ real-time MWL.

### 2.1. Participants

Twenty-one healthy Chinese male pilot cadets, with a mean age of 23.5 ± 3.5 years, participated in the study from the Flight Technology College of the Civil Aviation Flight University of China. They all held aviation commercial licenses from the Civil Aviation Administration of China (CAAC) and had logged an average of over 230 h of flight in both simulators and real aircraft. The Edinburgh Handedness Inventory revealed that all participants were right-handed. None of them had a history of mental illness or drug use, and all of them had normal or corrected-to-normal vision. The participants were instructed to have sufficient sleep and avoid consuming coffee or tea within 24 h before the experiment. They were also advised to have a good rest before the experiment and become familiar with the civil aviation simulator in advance of the formal experiment. All participants read and signed a consent form before the experiment and were later compensated for their participation. This research complies with the principles of the Declaration of Helsinki and has been approved by the Ethical Review Board of Southwest Jiaotong University (No. SWJTU-2109-001-QT).

### 2.2. Flight Simulator

A Cessna 172G1000 with an X-Plane flight simulation system was used for data collection (Figure 2). In this simulator, the corresponding internal instrument functions are realized. The display system simulates the real vision that the pilot sees in the flight process and matches it with the simulator model. Therefore, the pilots can perform more realistic flight tasks through the simulator and achieve a real−time dynamic interaction effect between the pilot and the virtual scene.

### 2.3. Wireless EEG Headset

Throughout the duration of the experiment, all subjects were required to wear EMOTIV INSIGHT 1.0—a 5-channel wireless EEG headset (WEEGH) produced by Emotiv Bioinformation Company. There are five semi-dry polymer sensors and two CMS/DRL references (left/right mastoid process alternatives). CMS (Common Mode Sense) is an active electrode and DRL (Driven Right Leg) is a passive electrode. Insight is non-invasive, user-friendly, durable, and portable. The sampling rate is 128 Hz and it also supports connection to mobile devices such as computers or mobile phones via Bluetooth 5.0. According to the international 10-20 EEG system, the electrode sensors are placed on the frontal (AF3, AF4), temporal (T7, T8), and occipital (Pz) lobes [32]. The details of EMOTIV INSIGHT 1.0 and its’ channel locations in different brain regions are shown in Figure 3.

### 2.4. Simulation Task and Procedure

All experimental procedures are modified based on airfield traffic pattern tasks, which were used as participants’ usual practice (Figure 4). Before the experiment, the subjects were informed in advance of the flight mission requirements and were instructed to perform appropriate flight exercises for practice to ensure that they were familiar with experimental procedures and tasks. Participants operated the Cessna 172G1000 aircraft with the cockpit view and took off from runway 19 of the Capital International Airport, executed a left turn, completed an airfield traffic pattern task, and finally landed on runway 19. The maximum bank of turning and rolling was restricted to 30° throughout the entire task. Autopilot was not to be used during the flight task, and the bank was not allowed to exceed 30°. Specific flight mission requirements are as follows:The aircraft starts taxiing along the runway. When the airspeed indicator reaches 55 knots, rotate the front wheels while maintaining a rate of climb (RoC) in upwind at 500 ft/min. When the altitude reaches 800 ft, participants can start to turn left;After the crosswind turn, maintain a track at 90° on the crosswind leg, and the distance of the crosswind leg should remain about 1.3 nautical miles;When the distance of the downwind leg exceeds the runway threshold by 1.3 nautical miles, the base leg begins. The altitude of the base leg should be maintained at 1100 ft. After the downwind leg, participants can begin to descend;The rate of approach should be maintained within 500 ft/min. And the altitude should be no more than 50 feet higher than the runway threshold when arrived.

Before the experiment, subjects were required to wear Emotiv Insight and to sit in front of the simulator. After that, the participants closed their eyes and rested for approximately five minutes. All participants followed the aircraft operation requirements until all tasks were completed. Following the experiment, the participants were required to recall their experiences and complete the NASA-TLX scales for each flight phase.

### 2.5. Survey

The NASA-TLX scale was used to evaluate the mental workload of pilots. Higher scores indicate a higher mental workload [33]. The NASA-TLX utilizes a 20-point visual analog score and provides an overall index of mental workload as well as the relative contributions of 6 subscales: mental, physical, temporal task demands, effort, frustration, and perceived performance [34]. Each subject was required to fill in the NASA-TLX scale immediately after completing a flight. The survey aims to mark the low, middle, and high MWL levels of the EEG data in different phases of flight missions.

### 2.6. EEG Processing

#### 2.6.1. EEG Pre-Processing

To prepare the EEG signals for analysis, several preprocessing steps were taken: (a) Channel location: The international 10–20 system was used for positioning channels; (b) Removing bad leads: channels with poor EEG signals due to channel faults or poor contact between electrodes and scalp were removed; (c) To reduce noise in EEG signals, band-pass filtering (0–40 Hz) was applied to the EEG signals; (d) Removing bad epochs: Epochs with an obvious artifact or bad signals were eliminated; (e) Independent component analysis (ICA): In this study, Electrooculogram (EOG) artifacts were removed according to EEG maps. All pre-processing was executed based on the open-source Python package MNE 1.3.0 in this study.

#### 2.6.2. Feature Extraction

To segment EEG signals from left to right, a series of orthogonal tapered window functions with a step size of 2 s and an overlap rate of 50% were employed. This resulted in obtaining *n*−1 time windows, where *n* represents the time length of the EEG signal in a flight phase (in seconds) [35]. For each time window, the fast Fourier transform (FFT) was used to obtain the frequency spectrum $f\left(k\right)$ of the EEG signal $f\left(n\right)$ using Equation (1), where *N* is the length of $f\left(n\right)$, and $W_{N}=\cos\frac{2\pi }{N}-i\sin\frac{2\pi }{N}$. Then, PSD can be approximated. This facilitated the analysis of specific frequency bands in EEG signals, which, in turn, can provide valuable insight into the mental workload status of pilots during flight tasks. Meanwhile, different features (Mean, variance, standard deviation, Peak-to-peak amplitude, skewness, kurtosis, root-mean-squared value, zero crossing, Hjorth mobility, and Hjorth complexity) from time-domain (TD) were also extracted. Furthermore, non-linear features (sample entropy and approximate entropy) were also extracted as supplements. Table 1 listed all features, and the 90 features were extracted in total ((PSD × 6 bands + 10 TD features + 2 entropy features) × 5 electrodes). To explain our study, we will refer to features other than PSD features as calculative features in the following sections.
(1)$$f\left[k\right]=\left\{\begin{array}{c} \overset{N=1}{\underset{n=0}{\sum }}f\left(n\right)W_{N}^{kn},\;0\leq k\leq N-1\\ 0,\;Otherwise \end{array}\right.$$

### 2.7. Feature Selection

To determine the effectiveness of various features, two distinct types of feature selection methods were employed, and their respective performances were compared. The first feature selection method, the K-W test, is utilized to ascertain the features that are significantly different from the others [39]. Another feature selection algorithm, SFFS, was also employed to analyze the performance of all features. The K-W test is suitable for comparing multiple independent groups without assuming equal variances or sample sizes, making it a valuable tool for researchers in a variety of fields [39]. SFFS’s ability to efficiently explore a large search space of possible feature subsets, its ability to handle highly correlated features, and its adaptability to various types of machine learning algorithms render it a valuable technique for improving the accuracy and efficiency of predictive models in diverse applications [40]. So, these two feature selection methods were conducted, and their performances were compared. The goal of feature selection is two-fold: to improve computational efficiency and to reduce the model’s generalization error by eliminating irrelevant features or noise. The motivation behind feature selection algorithms is to automatically identify a subset of features that are most relevant to the problem at hand. Ultimately, the optimal feature subset, based on both feature selection methods, was chosen as the input for all classifiers.

#### 2.7.1. Statistical Analysis

For the sake of ensuring appropriate statistical tests, a normality test was conducted on all features. If the characteristic parameters conform to the normal distribution, the T-test was used; otherwise, the Kruskal-Wallis (K-W) test was used. The normality test revealed that neither of the feature sequences followed a normal distribution, thus requiring the adoption of the K-W test. All statistical analyses were conducted using SPSS 26.0. Each EEG feature was individually tested using this method, and those with significant results were selected. To enhance the robustness of the selected features, z-scores were calculated to standardize the feature sets.

#### 2.7.2. Sequential Forward Floating Selection (SFFS)

SFFS can be considered an extension of a simpler sequential feature selection (SFS) algorithm [41]. Before explaining SFFS, let us first know SFFS. SFS algorithms are a family of greedy search algorithms that aim to reduce the dimensionality of an initial d-dimensional feature space to a k-dimensional feature subspace, where k < d. In brief, SFS removes or adds one feature at a time based on the classifier’s performance until a feature subset of the desired size k is achieved [40]. Compared with the SFS, the floating variant SFFS has an additional exclusion step to remove features once they are included so that a larger number of feature subset combinations can be sampled. It is important to emphasize that this step is conditional and only occurs if the resulting feature subset is assessed as “better” by the criterion function after the removal (or addition) of a particular feature. Furthermore, an optional check was added to skip the conditional exclusion steps if the algorithm gets stuck in cycles. Compared with SFS, SFFS is superior in any case in terms of univariate feature selection, where you only consider one feature at a time. For instance, assuming that there are correlations between some features or overfitting situations, SFFS is more flexible and better at considering feature interactions. A thorough illustration of SFFS is shown in Figure 5. In this study, the SFFS was wrapped with the KNN estimator, 10-fold cross-validation, and accuracy as the scoring parameter.

### 2.8. Machine Learning Algorithms

Six algorithms were adopted in each classification level to train the classifier to handle physiological data, respectively: support vector machine with a linear kernel (SVM-L), a radial basis function kernel (SVM-R), and a polynomial kernel (SVM-P), K-Nearest Neighbor (KNN), random forest (RF), and Bayesian neural network (BNN). The 10-fold cross-validation was performed to research the optimal parameters using a grid search.

(1) Support vector machine: The main goal of SVM is to resolve binary classification issues by searching for a hyperplane that can separate the two classes and optimizing the margin distance between the hyperplane and each class [42]. Assume that the t−th feature sample of the pilot is $x_{t}$ = $\left(x_{t1},x_{t2},\ldots ,x_{tm}\right)$, where  t=1,2,…,D and D is the sample capacities. Through nonlinear transformation, the result of $x_{t}$ is used to indicate the mental workload of pilots. Through nonlinear transformation, we can map $x_{t}$ to a high-dimensional linear space to convert a high-power problem to a quadratic programming problem. Radial Basis Function kernel (RBF) is $K\left(x_{t},x_{l}\right)=\exp\left(-\left(x_{t}-x_{l}\right)/\sigma^{2}\right)$ (σ is the width of the kernel function, t,l=1,2,…,D), Polynomial kernel function is $K\left(x_{t},x_{l}\right)={\left(x_{t}{}^{T}x_{i}\right)}^{d}$(d≥1 is the power of the Polynomial, degenerate to Linear kernel when d=1). Therefore, the decision function can be expressed as:(2)$$y\left(x\right)=\mathrm{sgn}\left(\overset{D}{\underset{t=1}{\sum }}y_{t}\alpha_{t}k\left(x_{t},x_{l}\right)+b\right)$$

For $x=\left(x_{1},x_{2,}\ldots ,x_{m}\right)$, which is a mental workload feature sample of a pilot to be identified.

(2) K-Nearest Neighbor: KNN is a non-parametric pattern recognition algorithm that identifies the classification of a new observation by comparing it to a training set of labeled observations, whereby it selects the k number of nearest points to the new data point and assigns the point to the class label that is most frequent among its k neighbors. It can also be used for regression analysis by calculating the average value of the K-nearest neighbors [6,43]. Assume that the t−th feature sample of pilots is $x_{i}$ = $\left(x_{1},x_{2,}\ldots ,x_{t}\right)$, where t=1,2,…,D, D is the sample volume, $y_{i}$ = ($y_{1},y_{2},\ldots ,y_{k}$), where  k=1,2,…,D, $y_{i}\in N^{*}$ is the category of the sample, where  i=1,2,…,n, and the training set of the input model is:$\;T=\left(x_{1},y_{1}\right),\left(x_{2},y_{2}\right),\ldots ,\left(x_{n},y_{n}\right)$ [44]. Euclidean distance is selected according to the distance measurement of the given point, which means $L\left(x_{i},x_{j}\right)={\left(\overset{n}{\underset{l=1}{\sum }}{\left|x_{i}^{\left(l\right)}-x_{j}^{\left(l\right)}\right|}^{2}\right)}^{\frac{1}{2}}$. Finding k points nearest to x in the training set T, covering the neighborhood of k points, which is recorded as $N_{k}\;\left(x\right)$, so the decision function y can be written as:(3)$$y=\arg\max_{y_{j}\underset{x_{i}\in N_{k}\left(x\right)}{\sum }I}(y_{i}=y_{j})$$
where i is the indicating function. According to the result, pilots’ MWL levels can be indicated. The hyperparameters of the number of neighbors were set to 1 after data tuning.

(3) Random forest: RF is a technique that combines multiple decision trees to solve classification problems. Each decision tree includes nodes that represent a conditional test, branches that indicate the outcome of a test, and leaves that represent one of the possible classes. [45]. Assume that the t−th feature sample of the pilot is $x_{i}=\left(x_{1},x_{2,}\ldots ,x_{t}\right)$, where  t=1,2,…,D, where D is the size of the sample. The bootstrap sampling method is used to conduct the t−th random sampling from the training set, collecting m times in total to obtain the sampling set $D^{T}$, and training the T−th decision tree model G with the sampling set $G_{T}\left(x\right)$. When training the decision tree model, randomly select some features from all the sample features on the node and select an optimal feature to divide the decision tree. Finally, the category with the most votes cast in the decision tree is taken as the final category to identify the flight phase [46]. The hyperparameters for the number of trees were set to 500 after data tuning.

(4) Bayesian neural network: BNN belongs to a class of neural networks that deal with model epistemic uncertainty. In a Bayesian framework, the posterior distribution of the model parameter ϖ is inferred when estimating a neural network structure $f^{\varpi }\left(.\right)$, which includes the number of layers, hidden units, and activation functions used, with ϖ representing the set of model parameters (neuron weights). BNN aims to assign a prior distribution $p\left(\omega \right)$ over the space of model parameters to represent a prior belief about each candidate model parameter. A likelihood function $p(Y_{i}|X_{i},\omega )(\omega \in \varpi )$ is then constructed to indicate the probability of generating the observed data [47]. By treating the parameters as random variables, BNN attempts to compute the posterior distribution p(ω,X|Y) of the parameters ω, as expressed in Equation (4). This distribution is used to obtain the predictive distribution by integrating over ϖ, as shown in Equation (5).
(4)$$p(\omega |X,Y)=\frac{p(Y|X,\omega )p\left(\omega \right)}{p(Y|X)}$$
(5)$$p\left(y*|x*,X,Y\right)=\int p\left(y*|x*,\omega \right)p(\omega |X,Y)d\omega$$

Performing Bayesian inference with the Markov chain Monte Carlo method to construct Bayesian neural networks is computationally intractable due to the nonlinearity and non-conjugacy of deep neural networks. To alleviate this computational burden, variational inference has been proposed, which approximates p(ω|X,Y) using a variational parametric distribution. However, this approach significantly increases the number of model parameters. To address this issue, the Monte Carlo dropout strategy has been added to construct a Bayesian neural network by dropping units and their incoming and outgoing connections when the corresponding binary variable is zero [48]. In contrast to BNNs, each parameter of the classical error backpropagation-based artificial neural network (ANN), which includes the input, hidden, and output layers, has a fixed value determined by training the network. The BNN of this study is constructed as a sequential neural network with three Bayesian layers, each activated by a ReLU function and followed by a dropout layer for regularization. The first layer takes in 90 input features and outputs 200 features, and the second layer takes in 200 features and outputs 200 features. The final layer takes in 200 features and outputs three classes. The prior mean and standard deviation of each linear layer are set to 0 and 0.1, respectively. The loss function is a combination of cross-entropy loss and Bayesian Kullback-Leibler (KL) divergence loss. The optimizer used is Adam with a learning rate of 0.001. A learning rate scheduler is also used to adjust the learning rate based on the performance of the model on the validation set. The model is trained for 10,000 epochs, with a KL weight of 0.05. It should be noted that traditional machine learning algorithms use features after dimension reduction as inputs, while deep learning algorithms such as BNNs directly use raw features, as the feature weighting concept is involved in the computing process.

### 2.9. Model Evaluation

The SVMs, KNN, RF, and BNN classification algorithms combined with 10-fold cross-validation were employed for EEG data classification. The accuracy, precision, recall, and *F*1 score were used to evaluate the classification performance, and defined by Equations (6)–(9):(6)$$accuracy=\frac{TP+TN}{TP+TN+FP+FN}$$
(7)$$precision=\frac{TP}{TP+FP}$$
(8)$$recall=\frac{TP}{TP+FN}$$
(9)$$F1=\frac{2*accuracy*recall}{accuracy+recall}$$

## 3. Results

### 3.1. Survey for Mental Workload Identification

The NASA-TLX scores in different flight phases of the airfield traffic pattern tasks from high to low are landing, take-off, and cruise for mental demand (74 ± 1.14 vs. 64 ± 1.29 vs. 46 ± 1.53), physical demand (43 ± 1.69 vs. 42 ± 1.56 vs. 41 ± 1.36), effort (52 ± 0.91 vs. 50 ± 1.33 vs. 45 ± 1.46) and frustration (36 ± 1.48 vs. 32 ± 1.40 vs. 25 ± 1.26). The perceived performance scores are opposite for landing, take-off, and cruise (64 ± 1.79 vs. 70 ± 1.32 vs. 78 ± 1.48), respectively. And the temporal task demand performance scores are influenced by the task design, which from high to low is cruise, take-off, and landing (51 ± 1.56 vs. 68 ± 1.31 vs. 64 ± 1.50). Furthermore, the average weighted score from all participants was 145.53 (±1.13) for landing, 105.6 (±1.29) for take-off, and 65.6 (±1.37) for cruise.

The MWL of different phases in airfield traffic pattern tasks were significantly different (*p* < 0.001) using one-way ANOVA. One-way ANOVA was used on NASA-TLX weighted scores in different flight phases (See Figure 6). The results show that the MWL levels between the three flight phases are significantly different. Besides, using an independent-sample *t*-test, it was found that the MWL of different phases in airfield traffic pattern tasks differed significantly (*p* < 0.001). After adjustment using the Bonferroni post-hoc test to account for three comparisons, the significance level was corrected at 0.017. The results showed that the MWL levels during different flight phases are significantly different (*p* < 0.017) (see Figure 6). For perceived MWL annotation, according to the results of statistical analyses, samples on landing were labeled as high MWL, samples on take-off were categorized as middle MWL, and samples on cruise were annotated as low MWL. Statistical analyses were performed using the Python package ‘scipy’ and ‘statannotations’.

### 3.2. Mental Workload-Related Neuro-Physiological Behaviors

Among all the extracted features, the power spectral density (PSD) is the most significant and widely used EEG indicator to describe brain unconscious activities during task execution [49]. Accordingly, we aimed to identify the PSD pattern associated with MWL variations for all participants. The most prominent PSD alterations, as depicted in Figure 7, indicate the brain area with the most significant changes in activity levels across different frequency spectra when participants experienced different MWL levels. Upon analysis of the data obtained, it was observed that under high and middle MWL conditions, clear variations in PSD were evident within the δ, θ, and α bands across all brain regions, except for the right side of the temporal lobe. Conversely, in low MWL situations, changes in PSD were only evident in the occipital lobe. With respect to the β1 bands, notable changes in PSD were observed in the right side of the temporal lobe under high MWL conditions, across all brain regions under middle MWL conditions, and only in the occipital lobe under low MWL conditions. Like β1, PSD changes in the β2 and γ bands were observed in the frontal lobe under low MWL conditions, while the changes in other regions were consistent with those observed in β1.

### 3.3. Feature Selection

This study employed two distinct feature selection methods: the K-W test (statistical method) and SFFS (machine learning algorithm). In addition, their performances were compared. For all data used for feature selection, z-scores were calculated for each sample to standardize them.

#### 3.3.1. Kruskal-Wallis Test

Initially, a normality test was conducted on all feature sets, and the results indicated that all samples did not follow a normal distribution. Consequently, the K-W test, which is commonly utilized for comparing two or more independent samples of equal or different sample sizes, was implemented [50]. The significance levels were established at *p* < 0.05 and *p* < 0.01, and the corresponding features were selected as input data, respectively. The analytical findings of all features are presented in Table 2. All statistical analyses were conducted utilizing SPSS 26.0.

#### 3.3.2. Sequential Forward Floating Selection

The SFFS algorithm, an improved greedy search algorithm based on SFS, was also performed to select features as input data to avoid the curse of dimensionality. For each feature set, 70% of the data were randomly selected for parameter tuning and model training, whereas 30% of the data were selected for model testing. The SFFS was wrapped with the KNN estimator with a 10-fold cross-validation method. The accuracy metric was used to evaluate the model. The performances of each feature are shown in Figure 8 and Figure 9, and the features achieving the best performance are selected as different inputs feeding into different classifiers. It should be noted that PSD is an independent input into SFFS. The reason is that PSD is the most significant and effective feature [49], and other features are supplements.

### 3.4. Performance Comparison

To develop an effective system for detecting pilots’ real-time MWL and further examine the effectiveness of this system, the authors compared the performances of various algorithms, the performances of the two feature selection methods, and the contributions of EEG data from different brain regions. For each feature set, 70% of the data were randomly selected to tune the hyperparameters and train the model, and 30% of the data were selected to test the model. And all classifiers were performed using the Python packages “sklearn” and “torchbnn”. Meanwhile, multiclass classifiers combined with 10-fold cross-validation were used. To systematically evaluate the model, the mean values of all metrics over the 10 runs and their standard deviations for all indicators were calculated to estimate the generalizability of the various machine learning models.

### 3.5. Performance Comparison of Various Classifiers

In order to evaluate the performance of different classifiers and train the optimal parameters, the performances of five machine learning classifiers (SVM-L, SVM-R, SVM-P, KNN, and RF) were compared first. Two feature sets (30 PSD features and 60 calculative features) were used as input for all classifiers, respectively. The detection indicators used to evaluate the performance were accuracy, precision, recall, and F1 score, as presented in Table 3. Based on the results in Table 3, it is apparent that both KNN and RF classifiers outperformed SVMs. This is because SVMs are more effective in binary classification tasks; however, this study involved a multi-class classification problem. Consequently, to reduce computational costs and eliminate redundant calculations, only KNN and RF classifiers were chosen for further study.

#### 3.5.1. Performance Comparison of Feature Selection Algorithms

Previous research has shown that redundant features can negatively impact model computational speed and accuracy [51]. Therefore, this section compares different feature sets selected using the K-W test and SFFS algorithm to identify the feature set that provides the best computational performance and fastest speed as the input to the model. We also compare its performance with that of BNN. As described in Section 3.3, for the K-W test, PSD features with *p* < 0.01 and *p* < 0.05 were selected as input data 1 and input data 2, respectively. Additionally, calculative features with *p* < 0.01 and *p* < 0.05 were selected as input data 3 and input data 4, respectively. For SFFS, the feature set that resulted in the best model performance, including all features, was selected as the input. Among the PSD features, two feature sets, containing 5 and 13 features with similar performance (see Figure 6), were both selected as input data 5 and input data 6, respectively. For calculative features, 23 features were selected as input data. Table 4 displays the features contained in each feature set, while Table 5 presents the performance of each feature set. In Table 5, the results of PSD feature sets are bold. Based on the results in Table 5, two conclusions can be drawn: (a) the best performance appeared in feature sets that included the most features, and (b) although the performance of the feature sets selected by SFFS is worse (85.51%) compared to the feature sets selected by K-W test (87.30%), input data 6 (containing 13 features) has fewer features compared to input data 2 (containing 22 features).

To validate and elucidate the conclusion stated in the previous paragraph and identify the optimal model, this study incorporated additional feature sets for further investigation to compare their performances: all Power Spectral Density (PSD) features (Input data 8), all PSD features combined with input data 4 in Table 4 (Input data 9), and all features (Input data 10). The outcomes are displayed in Table 6. Moreover, the performance of the Bayesian Neural Network (BNN) was assessed and compared, yielding accuracy, precision, recall, and F1 score values of 70.67%, 75.22%, 70.59%, and 70.80%, respectively. Intriguingly, based on the findings presented in Table 5 and Table 6, it was concluded that the best performance was attained using all PSD features (input data 8) as input. Furthermore, the highest values in accuracy (87.57%), recall (72.76%), and F1 score (70.57%) were achieved by the K-Nearest Neighbors (KNN) algorithm, while the Random Forest (RF) classifier obtained the highest value in precision (88.35%).

#### 3.5.2. The Impact of EEG Features from Different Brain Regions on Mental Workload Detecting

For the real-time mental workload detection systems developed in this study, the type of inputted EEG features would be a key factor in their performance as well as their feasibility. According to all comparison results, feeding the data that contains all PSD features acquired by WEEGH into the KNN classifier can achieve the highest performance. Therefore, to further examine the contributions of PSD features from different brain regions, different bands (δ, θ, α, β1, β2, γ) of AF4 and AF3 (frontal lobe), T7 and T8 (temporal lobe), and Pz (occipital lobe) were fed into the KNN classifier for comparison. Based on the findings presented in Table 7, utilizing mixed EEG features as inputs yields superior performance compared to utilizing EEG features from a specific band. Additionally, the ranking according to the accuracy of different regions is temporal, frontal, and occipital, from strong to weak, based on this system.

## 4. Discussion

In this study, 21 pilots with more than 230 h in both simulators and real aircraft were recruited, and a total of more than 8700 effective samples were collected. NASA-TLX scales were first calculated with the purpose of labeling samples as different MWLs. Meanwhile, the K-W test and SFFS were used to select the best input among PSD features and calculative features, and then the performances of five machine learning algorithms and a Bayesian neural network were compared. Using a 10-class multi-class KNN classifier wrapped with 10-fold cross-validation can achieve the best average accuracy of 87.57% (see Table 6). Yiu et al. employed a Bayesian neural network-based multi-classification algorithm to identify various weather conditions, achieving an accuracy of 66.5% [22]. Another study used a stacking ensemble algorithm with an EEG device containing 20 electrodes to distinguish between different levels of pilot mental workload, achieving an accuracy of 91.67% [52]. However, this study only recruited 10 pilots, which may limit the generalizability of its findings. Both studies employed multi-channel EEG devices with 32 and 20 channels, respectively. To further examine the physiological patterns and contributions of different brain regions based on this real-time MWL detecting system, the PSD changes of brain activities (see Figure 6) and the performances of EEG features from different bands of different brain regions were also researched (see Table 7). All results of this study can prove that this real-time MWL detecting system is effective in pilots, providing insight for developing a human-centered MWL detecting interactive system in real aviation flights as well as contributing to EEG research under real flight conditions.

### 4.1. The Effectiveness of the Survey and the Experiment

To scientifically develop and validate the system for detecting pilots’ MWL, an experiment that included various MWL levels based on airfield traffic pattern tasks was designed. According to the acceptable high MWL threshold [53], task loads in the take-off, cruise, and landing phases have induced a sense of middle, low, and high workloads within participants, respectively, achieving a NATA-TLX score of 105.6 (±1.29), 65.6 (±1.37), and 145.53 (±1.13). This result is consistent with the judgments of flight instructors and experimental pilots. Therefore, this experiment, which included three MWL levels, can be used to develop and validate this detection system. In addition, the collected NASA-TLX scores were also used to mark the samples’ MWL. Consequently, multi-class classifiers were selected for training this model.

### 4.2. Interpreting Physiological Patterns Related to Mental Workload Changes

Although this system is based on a consumer-grade WEEGH, the physiological patterns with MWL changes were also clearly revealed based on brain activity. The temporal lobes are believed to play an important role in processing affect and emotions, language, and certain aspects of visual perception, while the frontal lobe plays an integral role in higher-level cognitive processes such as attention and working memory [18]. Felix et al. found peak performers decrease δ waves when high focus and peak performance are required; the θ band is related to short-term memory [54], and the α band is associated with relaxing and healing. It is worth noting that there was no evident participation in the right temporal lobe among both, δ, θ, and ɑ bands based on this system. This result is consistent with previous findings pointing out that the dominant temporal lobe, which is involved in understanding language and learning and remembering verbal information, is on the left side in most people [55]. The neurophysiological results indicate that a decrease in the power of the δ and α frequency bands in the temporal lobe corresponds to high awareness, deep thinking, and increased workload execution. Additionally, a decrease in the power of the θ frequency band suggests that the tasks were not new for the participants. According to Dasari et al.’s work, β1 usually means a normal workload level, while an increase in 2 and power is associated with more mental effort [15]. From our results, we can see that the low MWL task was mainly associated with task execution, but the middle and high MWL tasks needed more information handleability. To investigate the impact of EEG features on the system, we fed the EEG features from different bands in different brain regions into a multi-class KNN model individually. Comparison results are presented in Table 7. Although the performance ranking of different bands across multiple brain regions fluctuates, the mixed EEG bands perform the best. This is because there are functional connections between each brain region. Regarding the contribution of different brain regions to this system, the ranking from highest to lowest is temporal, frontal, and occipital. The reason is probably that there is only one electrode on the occipital lobe. Considering all neuro-physiological results (Section 3.2), this study can not only achieve good performance but also reveal physiological patterns with MWL changes.

### 4.3. Explaining the Works in Developing the Real-Time Mental Workload Detecting System

The primary objective of Section 3.3 and Section 3.4 is to evaluate the performance of various classifiers and feature selection methods in the context of physiological pattern analysis to aid in developing a real-time MWL detection system. This research utilizes two feature selection algorithms for the following reasons: (a) to assess the effectiveness and performance of the MWL detecting system using pilot EEG data collected by the WEEGH, and (b) to determine the presence of redundant features or the need for additional features. As described in Table 3 and Table 4, multiple features were selected and combined into distinct feature sets, with their performances compared in Section 3.4. The results indicate that both the K-W test and the SFFS method did not contribute to performance improvement, likely due to the absence of redundant features in the original PSD features. Conversely, this finding also corroborates the effectiveness and high quality of the EEG data acquired by the WEEGH in detecting pilots’ MWL. Following the performance comparison of various feature sets, it was determined that the optimal input consists of all PSD features, which, when used with a multi-class KNN classifier, achieves an accuracy of 87.57%. These findings support the development of the MWL-detecting system and confirm its feasibility.

### 4.4. Practical Applications

A possible online real-time pilots’ MWL detecting system is illustrated in Figure 10. With the development of semi-dry sensor technologies and wireless EEG acquisition technologies [24], some devices in the consumer category that measure activity from all cortical lobes of the brain were produced. This WEEGH can provide in-depth information usually found in research devices [32,56]. Thus, researchers can create custom montages to integrate wireless electrodes into pilots’ headphones for seamless, non-invasive, and accurate EEG detection. The EEG data can be transmitted to the processing unit through Bluetooth. Then the pilots’ MWL can be detected and estimated in real-time by the administrators. Meanwhile, an adaptive MWL evaluation model can be determined according to the environment and management requirements, including airway conditions, flight speed, weather conditions, etc. If the MWL of pilots is judged irregularly by administrators using the evaluation model, the feedback intervention strategies (e.g., seat vibration, visual and audio warnings) will be triggered to wake up the pilots’ vigilance. Other applications could also be developed based on the proposed method in this study to contribute to pilots’ operational safety.

This study presents a real-time pilot mental workload (MWL) detection system utilizing a consumer-grade wireless electroencephalogram headset (WEEGH) with EEG features. The optimal model, KNN with all power spectral density (PSD) features, was determined by comparing and training the performance of various algorithms and selecting features based on different feature selection algorithms. The feature selection results demonstrated that the EEG data collected by this device is of high quality and accurate for detecting pilots’ MWL. Furthermore, PSD alterations were further examined to confirm the system’s ability to detect pilots’ MWL, with results indicating that pilots’ PSD variations are captured by the system. The impact of PSD features from different brain regions and bands revealed that PSD features extracted from electrodes in the temporal region contribute significantly to this system, which can inform the design of portable MWL detecting systems for aircraft. A key advantage of the proposed system is the wireless and portable EEG device, with its performance verified, suggesting the potential for developing more systems based on wireless EEG headsets for various real-world driving scenarios, such as aviation, rail transportation, and shipping. Another strength of our system is the replaceable model, which allows for continuous development and improvement, with more advanced models potentially providing better detection performance.

Despite the study’s success in identifying and detecting MWL levels during simulated flight driving with a WEEGH, some limitations are acknowledged. While the feasibility of detecting pilots’ real-time MWL with EMOTIV INSIGHT during simulated aircraft driving is demonstrated, further research with larger-scale experienced pilots is needed to draw general conclusions and build robust systems. Additionally, considering the substantial variation in brain activity between individuals, more adaptive and powerful models should be explored to achieve higher accuracy. Participants may also be more relaxed in simulator conditions, and few existing studies have examined the differences in brain activity between simulated and real situations. Data obtained in actual working conditions would be more authentic and representative, but annotating datasets collected through self-reporting methods during real aircraft driving is challenging. Future work will focus on actual working conditions using our proposed system to investigate how to train the model for detecting pilots’ mental workload with an unlabeled dataset. Possible approaches include training a model in the lab and applying it to real-world problems or employing active learning or semi-supervised learning methods. Additionally, efforts to improve model performance and achieve higher classification accuracy will be investigated.

## 5. Conclusions

This paper incorporates advanced algorithms and semi-dry sensor technologies from the fields of artificial intelligence and EEG acquisition as references to propose an innovative real-time system based on a wireless EEG headset (WEEGH) for identifying and detecting pilots’ real-time mental workload (MWL). Machine learning algorithms and Bayesian neural networks were evaluated for enhancing classification accuracy, and two feature selection methods were employed to identify the most effective features for MWL classification. The results demonstrated that the highest accuracy, 87.57%, could be achieved using a multi-class KNN classifier with all power spectral density (PSD) features as inputs. To verify the effectiveness of the proposed system, EEG data were collected from 21 pilots in a simulated experiment. The results indicated that the proposed system exhibited high classification accuracy, meeting the requirements for real-time detection of pilots’ MWL. Additionally, PSD variations in different brain regions and PSD features from different brain regions and frequency bands were investigated, confirming that the data collected by the wireless EEG headset also held significant research value and provided in-depth information on pilots’ brain activity typically found in research devices. Consequently, numerous systems based on wireless EEG headsets can be developed for application in various real-world driving scenarios, and the present system contributes to future studies in actual flight conditions.

## Figures and Tables

**Figure 1 entropy-25-01035-f001:**
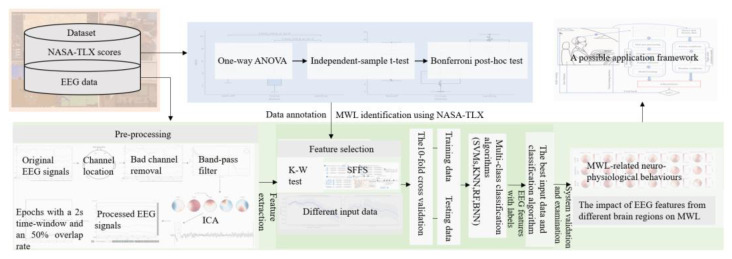
The framework of the proposed system and its potential application.

**Figure 2 entropy-25-01035-f002:**
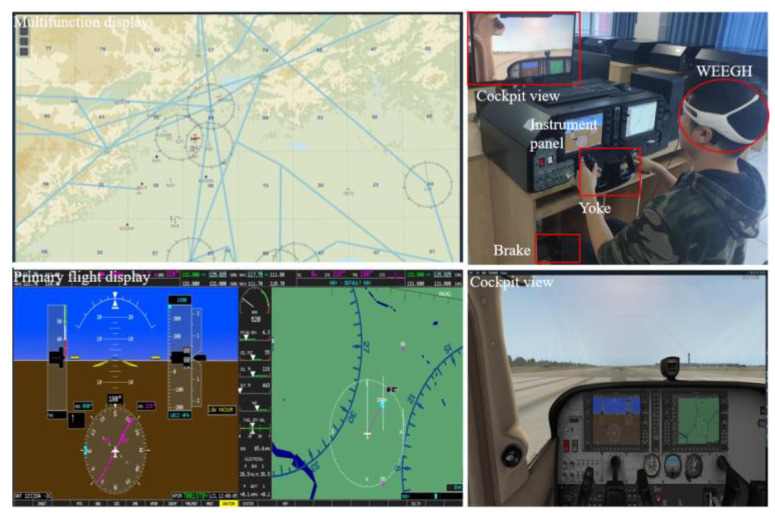
Cessna 172G1000 simulator.

**Figure 3 entropy-25-01035-f003:**
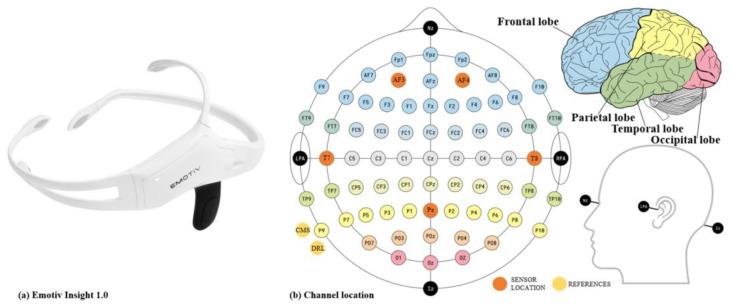
EMOTIV INSIGHT 1.0 (**a**) and its channel location of 10–20 system (**b**).

**Figure 4 entropy-25-01035-f004:**
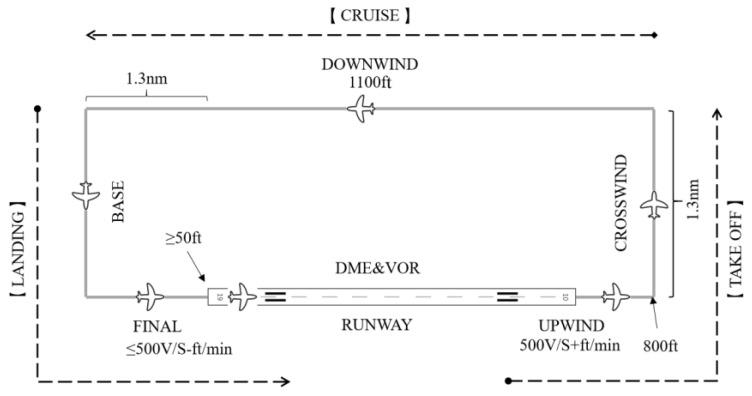
Modified airfield traffic pattern tasks.

**Figure 5 entropy-25-01035-f005:**
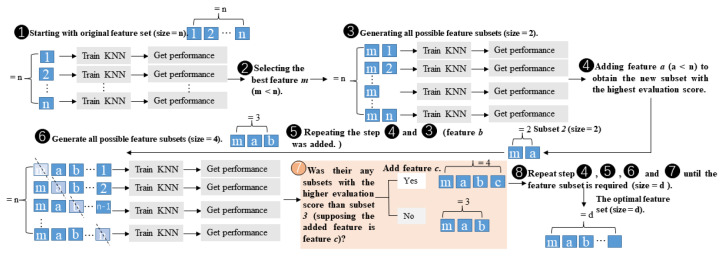
The illustration of SFFS (step 7 is the floating round of SFFS and the other steps are regular rounds of SFS).

**Figure 6 entropy-25-01035-f006:**
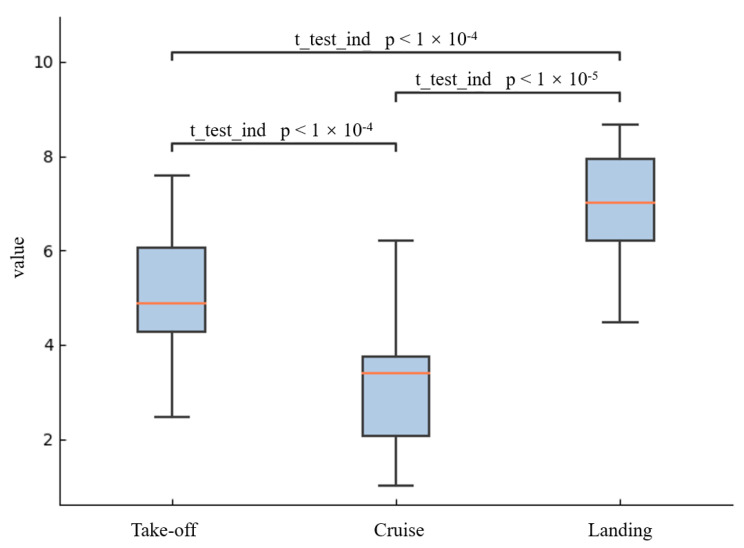
Statistical analysis result of NASA-TLX scores.

**Figure 7 entropy-25-01035-f007:**
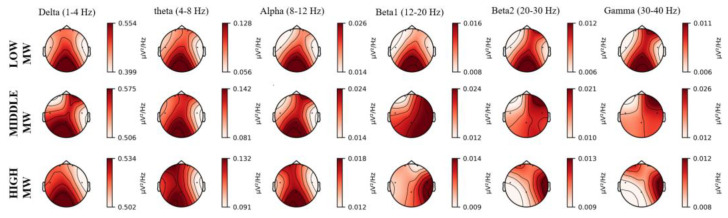
The most dominant features from all participants represent the areas of PSD’s largest changes in brain activities under different MWL conditions.

**Figure 8 entropy-25-01035-f008:**
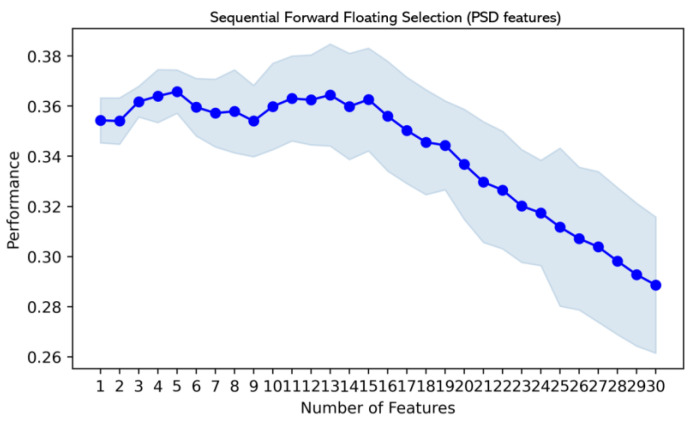
The performances of PSD features.

**Figure 9 entropy-25-01035-f009:**
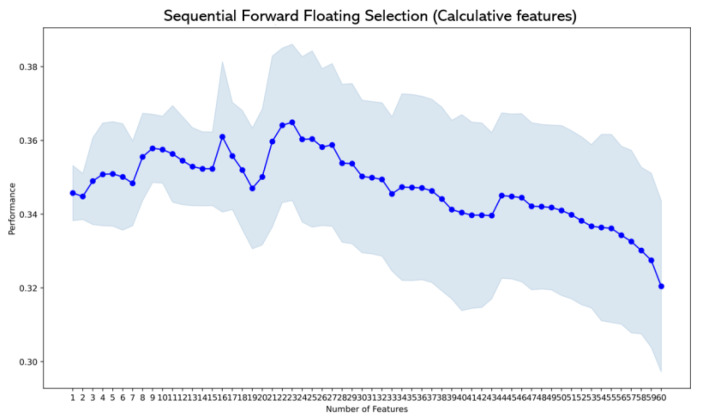
The performances of calculative features.

**Figure 10 entropy-25-01035-f010:**
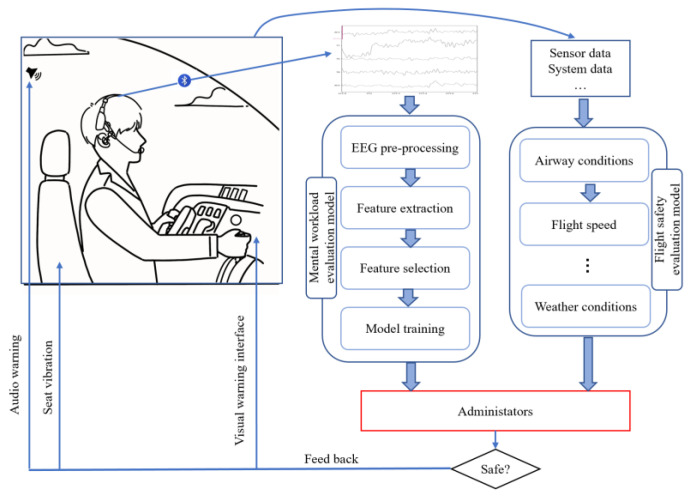
A possible application framework for pilots’ real-time mental workload evaluation.

**Table 1 entropy-25-01035-t001:** Features of EEG data used in detecting mental workload.

Domain	Features	Unit	Description
Frequency	Power spectral density (PSD)	μv^2^/Hz	power spectral density of oscillations at a specific frequency band in a particular brain region of a participant during a single trial.
Time	Mean	μv	a measure of central tendency and provides information about the overall level of activity in the EEG signal.
	Variance (vari)	μv^2^	providing information about the spread of the data and used to measure signal variability in EEG analysis.
	Standard deviation (std)	μv	providing information about the degree of variability in the data, it is used to measure signal variability in EEG analysis.
	Peak–to–peak amplitude (ptp-amp)	μv	a measure of the difference between the highest and lowest points of a single waveform in the EEG signal.
	Skewness	-	a measure of the asymmetry of the probability distribution of recorded EEG data about its mean.
	Kurtosis	-	a measure of the “tailedness” of the probability distribution of recorded EEG data.
	Root—mean squared (rms) value	μv	the root—mean squared value of recorded EEG data.
	Zero crossing (zc)	-	a point where the EEG signal changes sign in its waveform.
	Hjorth mobility (hm)	μv/Hz	represents the mean frequency or the proportion of standard deviation of the power spectrum [36].
	Hjorth complexity (hc)	-	indicates how similar the shape of a signal is to a pure sine wave, and gives an estimation of the bandwidth of the signal [36].
Non-linear	Sample entropy (se)	-	used for assessing the complexity of EEG time-series signals [37].
	Approximate entropy (ae)	-	a technique used to quantify the amount of regularity and the unpredictability of fluctuations over time-series data [38].

**Table 2 entropy-25-01035-t002:** The result of the Kruskal-Wallis test.

Features	*p* (AF4)	*p* (AF3)	*p* (T7)	*p* (T8)	*p* (Pz)
δ	0.000 **	0.000 **	0.002 **	0.002 **	0.192
θ	0.004 **	0.005 **	0.015 *	0.023 *	0.396
α	0.000 **	0.236	0.194	0.401	0.000 **
β1	0.000 **	0.008 **	0.223	0.257	0.000 **
β2	0.000 **	0.000 **	0.004 **	0.172	0.000 **
γ	0.000 **	0.000 **	0.002 **	0.029 *	0.000 **
Mean	0.860	0.887	0.502	0.877	0.930
Variance	0.000 **	0.000 **	0.008 **	0.252	0.392
Standard deviation	0.000 **	0.000 **	0.008 **	0.252	0.392
Peak–to–peak amplitude	0.000 **	0.001 **	0.009 **	0.426	0.377
Skewness	0.741	0.000 **	0.412	0.996	0.080
Kurtosis	0.713	0.780	0.185	0.004 **	0.343
Root–mean squared value	0.000 **	0.001 **	0.000 **	0.209	0.187
Zero crossing	0.008 **	0.000 **	0.005 **	0.144	0.053
Hjorth mobility	0.557	0.000 **	0.024 *	0.001 **	0.038 *
Hjorth complexity	0.323	0.000 **	0.003 **	0.000 **	0.147
Sample entropy	0.784	0.000 **	0.043 *	0.000 **	0.020 *
Approximate entropy	0.493	0.000 **	0.007 **	0.000 **	0.008 **

Statistically significant effects (*p* < 0.05) are denoted by *, and statistically significant effects (*p* < 0.01) are denoted by **.

**Table 3 entropy-25-01035-t003:** Comparison of classifier results.

Performance	PSD Features					Calculative Features				
Classifiers	KNN	RF	SVM-L	SVM-R	SVM-P	KNN	RF	SVM-L	SVM-R	SVM-P
Accuracy	87.57 ± 0.72	86.52 ± 0.95	66.48 ± 1.22	71.35 ± 1.17	67.85 ± 1.19	79.55 ± 0.86	82.56 ± 1.33	64.59 ± 1.21	70.43 ± 1.41	68.20 ± 1.26
Precision	80.47 ± 0.92	88.35 ± 1.85	22.16 ± 0.41	58.11 ± 14.67	55.53 ± 12.25	72.43 ± 0.97	86.64 ± 1.79	22.24 ± 0.76	77.46 ±17.03	68.66 ± 13.38
Recall	72.76 ± 1.40	65.94 ± 1.87	33.30 ± 0.01	41.22 ± 1.01	35.37 ± 0.44	65.40 ± 1.01	59.78 ± 1.98	33.30 ± 0.01	44.34 ± 1.29	40.79 ± 0.64
F1 score	75.73 ± 1.21	70.57 ± 2.28	26.62 ± 0.29	40.56 ± 1.45	30.96 ± 0.76	67.09 ± 1.43	65.18 ± 3.01	26.67 ± 0.33	46.35 ± 1.84	40.87 ± 1.18

**Table 4 entropy-25-01035-t004:** Contained features in various feature sets.

Feature Sets	Contained Features
Input data 1	AF4-δ, AF4-θ, AF4-α, AF4-β1, AF4-β2, AF4-γ, AF3-δ, AF3-θ, AF3-β1, AF3-β2, AF3-γ, T7-δ, T7-β2, T7-γ, T8-δ, Pz-α, Pz-β1, Pz-β2, Pz-γ
Input data 2	AF4-δ, AF4-θ, AF4-α, AF4-β1, AF4-β2, AF4-γ, AF3-δ, AF3-θ, AF3-β1, AF3-β2, AF3-γ, T7-δ, T7-θ, T7-β2, T7-γ, T8-δ, T8-θ, T8-γ, Pz-α, Pz-β1, Pz-β2, Pz-γ
Input data 3	AF4-vari, AF4-std, AF4-ptp-amp, AF4-rms, AF4-ae, AF3-vari, AF3-std, AF3-ptp-amp, AF3-skewness, AF3-rms, AF3-ae, AF3-se, AF-3zc, AF3-hm, AF3-hc, T7-vari, T7-std, T7-ptp-amp, T7-rms, T7-ae, T7-se, T7-zc, T7-hm, T7-hc, T8-kurtosis, T8-se, T8-zc, T8-hm, T8-hc, Pz-hm, Pz-hc
Input data 4	Input data 3 and Pz-skewness and Pz-se
Input data 5	AF4-α, AF4-β2, T7-δ, Pz-α, Pz-β1
Input data 6	AF4-δ, AF4-β1, AF4-γ, AF3-δ, AF3-α, AF3-β1, AF3-β2, AF3-γ, Pz-θ, Pz-α, Pz-β1, Pz-β2, Pz-γ
Input data 7	AF4-vari, AF4-std, AF4-ptp-amp, AF4-rms, AF4-ae, AF3-vari, AF3-std, AF3-ptp-amp, AF3-skewness, AF3-rms, AF3-ae, AF3-se, AF3-zc, AF3-hm, AF3-hc, T7-vari, T7-std, T7-ptp-amp, T7-rms, T7-ae, T7-se, T7-zc, T7-hm, T7-hc, T8-se, T8-zc, T8-hm, T8-hc, Pz-hm, Pz-hc
Input data 8	All PSD features
Input data 9	All features

**Table 5 entropy-25-01035-t005:** The performance comparison of two feature selection methods.

Performance	Algorithm	Input Data 1	Input Data 2	Input Data 3	Input Data 4	Input Data 5	Input Data 6	Input Data 7
Accuracy	KNN	**85.50 ± 0.56**	**86.3 ± 0.9**	80.52 ± 2.15	80.11 ± 2.21	**83.15 ± 0.65**	**85.01 ± 0.81**	79.46 ± 1.58
	RF	**85.99 ± 0.61**	**87.3 ± 0.5 ***	81.54 ± 0.99	81.86 ± 1.02 *	**83.54 ± 0.52**	**84.85 ± 0.69**	81.29 ± 0.71
Precision	KNN	**79.24 ± 2.21**	**80.8 ± 1.0**	70.47 ± 3.09	70.04 ± 4.02	**72.85 ± 2.11**	**75.48 ± 1.46**	68.60 ± 2.72
	RF	**87.92 ± 1.56**	**88.5 ± 1.6 ***	85.18 ± 2.54	86.27 ± 1.19 *	**84.08 ± 1.35**	**87.86 ± 1.45**	85.08 ± 2.34
Recall	KNN	**72.08 ± 1.71**	**72.36 ± 0.89 ***	61.79 ± 4.26	62.98 ± 4.24 *	**67.75 ± 1.29**	**69.21 ± 1.70**	61.58 ± 1.33
	RF	**66.03 ± 1.81**	**66.57 ± 1.05**	58.35 ± 1.67	59.05 ± 1.10	**63.62 ± 1.28**	**63.42 ± 1.14**	57.83 ± 1.19
F1 score	KNN	**74.83 ± 1.75**	**75.20 ± 0.95 ***	64.45 ± 4.47	64.69 ± 4.41 *	**69.84 ± 1.51**	**71.64 ± 1.66**	63.97 ± 1.61
	RF	**70.67 ± 2.36**	**71.35 ± 1.24**	63.06 ± 2.19	64.06 ± 1.27	**68.45 ± 1.57**	**67.87 ± 1.60**	62.65 ± 1.64

The best performances of PSD feature sets (bold) and calculative feature sets were both denoted by *.

**Table 6 entropy-25-01035-t006:** The supplementary performance comparison.

Performance	Algorithm	Input Data 8	Input Data 9	Input Data 10
Accuracy	KNN	87.57 ± 0.72 *	82.80 ± 2.20	81.34 ± 2.15
	RF	86.52 ± 0.95	83.70 ± 0.68	82.98 ± 0.93
Precision	KNN	80.47 ± 0.92	76.30 ± 3.33	72.44 ± 5.39
	RF	88.35 ± 1.85 *	87.60 ± 1.67	88.20 ± 1.43
Recall	KNN	72.76 ± 1.40 *	67.86 ± 1.90	65.67 ± 1.34
	RF	65.94 ± 1.87	62.15 ± 1.66	60.50 ± 1.38
F1	KNN	75.73 ± 1.21 *	70.76 ± 2.02	67.82 ± 2.63
	RF	70.57 ± 2.28	67.41 ± 2.12	65.66 ± 1.87

The best performance was denoted by *.

**Table 7 entropy-25-01035-t007:** The performance of the KNN classifier when using EEG features from different bands of various brain regions.

Region	Inputs	Accuracy	Precision	Recall	F1 Score
Frontal (AF4, AF3)	δ	80.49 ± 0.38	66.82 ± 0.94	62.74 ± 0.80	64.42 ± 0.78
	θ	80.03 ± 0.88	65.94 ± 1.63	61.40 ± 0.94	63.16 ± 1.15
	α	79.88 ± 0.78	65.70 ± 1.69	61.06 ± 0.98	62.93 ± 1.22
	β1	79.31 ± 1.06	65.17 ± 1.86	60.52 ± 1.70	62.32 ± 1.72
	β2	80.24 ± 0.79	66.91 ± 1.38	61.90 ± 1.15	63.83 ± 1.12
	γ	79.40 ± 0.81	65.60 ± 2.08	60.96 ± 1.29	62.76 ± 1.53
	mix	84.35 ± 0.85	74.91 ± 1.71	68.17 ± 1.53	70.74 ± 1.58
Temporal (T7, T8)	δ	80.97 ± 0.61	68.46 ± 1.44	63.31 ± 1.14	65.32 ± 1.22
	θ	80.86 ± 0.43	68.71 ± 0.61	63.68 ± 0.93	65.64 ± 0.64
	α	79.52 ± 0.93	66.87 ± 2.29	61.08 ± 1.18	63.19 ± 1.52
	β1	81.20 ± 0.78	67.84 ± 1.75	63.27 ± 1.19	65.06 ± 1.26
	β2	81.59 ± 0.71	69.40 ± 1.63	64.41 ± 1.39	66.36 ± 1.39
	γ	81.65 ± 0.69	69.49 ± 1.20	64.47 ± 1.56	66.45 ± 1.45
	mix	86.20 ± 0.70	77.38 ± 1.47	71.04 ± 1.01	73.49 ± 0.97
Occipital (Pz)	δ	79.25 ± 0.79	65.61 ± 2.33	61.01 ± 1.38	62.82 ± 1.69
	θ	79.14 ± 1.03	64.85 ± 2.18	60.00 ± 1.81	61.84 ± 1.91
	α	79.76 ± 0.89	65.75 ± 1.42	61.77 ± 1.51	63.39 ± 1.50
	β1	79.59 ± 0.82	65.68 ± 2.51	61.27 ± 1.67	62.97 ± 1.93
	β2	79.00 ± 0.95	63.57 ± 1.51	59.76 ± 1.26	61.28 ± 1.34
	γ	79.09 ± 1.01	64.72 ± 2.12	60.28 ± 1.20	61.99 ± 1.40
	mix	82.38 ± 1.22	69.78 ± 2.58	65.48 ± 1.91	67.24 ± 2.12
All five electrodes	δ	82.95 ± 0.78	71.93 ± 2.89	66.40 ± 2.01	68.60 ± 2.30
	θ	82.31 ± 0.83	70.66 ± 2.63	65.25 ± 1.51	67.37 ± 1.82
	α	81.66 ± 1.14	69.81 ± 1.73	64.34 ± 1.46	66.44 ± 1.42
	β1	82.43 ± 0.71	70.08 ± 1.91	64.78 ± 1.50	66.83 ± 1.50
	β2	83.70 ± 0.59	79.53 ± 1.32	64.48 ± 0.73	68.54 ± 0.99
	γ	83.48 ± 0.85	72.81 ± 2.30	66.81 ± 1.37	69.19 ± 1.66
	mix	87.63 ± 0.68	80.39 ± 1.53	73.88 ± 1.49	76.51 ± 1.43

## Data Availability

Data is unavailable due to privacy.

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
