# Peer review of "Detection of Pilot’s Mental Workload Using a Wireless EEG Headset in Airfield Traffic Pattern Tasks"

_entropy, 2023, doi:10.3390/e25071035_

Round 1

Reviewer 1 Report

The content of the paper is very interesting and therefore the paper is worth publishing. However, before publication serious changes should be made to increase its readability. The problem with the paper is that it contains a lot of additional information (only loosely related to the merit of the paper) that is not necessary to understand the paper's message. This information should be traced and removed since it distracts the reader from the paper core. 

Moreover, some concepts also repeat in different parts of the paper or are divided between different sections; for example, information on EEG origin/content/interpretation should be kept in one place (e.g. in Introduction) but is scattered all over the paper. Other pieces of information are just not in the right place; for example, the general description of the feature selection problem should appear in 2.7, not in 2.7.2. It sounds like many people wrote different parts of the paper but none of them read the whole content to make the final adjustment.

There are also some language mistakes, which sometimes hinder the understanding of the paper content (especially in figures and tables).

One more issue is with the State of the Art. While a lot of broad aspects are commented on in Introduction, there are almost no references (in Introduction) directing the reader to papers specifically on EEG features/locations/bands/indicators used by other researchers for MWL analysis.

Most of these remarks do not touch the merit of the content which in my opinion is valuable. Their aim is to make the paper more understandable and easier to follow. Summing up, the paper should be significantly shortened (especially the Introduction) and more attention should be put on the merit of the survey.

Reviewer 2 Report

The subject addressed in this work is not new. In the specialized literature, at least in recent years, other works have been published on this topic, with even much better results (for example the paper: Using machine learning methods and EEG to discriminate aircraft pilot cognitive workload during fight)

On the other hand, there is no comparison of the results of the work with those in the literature. A comparison with other results is essential, especially if the topic addressed is not new. In the present case, the theme is not new, or revolutionary, neither from the point of view of the paradigm nor from the point of view of the signal processing or classifier methods used, to be presented only at the level of the obtained results.

The flight description part is much too detailed compared to the signal processing part (eg data normalization, feature selection, why the Kruskal-Wallis (K-W) test was chosen, and other data or signal processing details).

Why was not used EEG data already existing in the literature on this field? In this way, the results could be compared to precisely the same recordings.

At  http://iam.cvc.uab.es/portfolio/e-pilots-dataset/ 

there are EEG and ECG signals data collected from the pilots.

Round 2

Reviewer 1 Report

I do not have any further comments.

Reviewer 2 Report

The part of comparing the results with the existing ones can be improved.